# Effect of Target Semantic Consistency in Different Sequence Positions and Processing Modes on T2 Recognition: Integration and Suppression Based on Cross-Modal Processing

**DOI:** 10.3390/brainsci13020340

**Published:** 2023-02-16

**Authors:** Haoping Yang, Chunlin Yue, Cenyi Wang, Aijun Wang, Zonghao Zhang, Li Luo

**Affiliations:** 1School of Physical Education and Sports Science, Soochow University, Suzhou 215021, China; 2School of Education, Soochow University, Suzhou 215023, China

**Keywords:** attentional blink, selective attention, exogenous attention, endogenous attention, cross-modal integration, semantic congruency

## Abstract

In the rapid serial visual presentation (RSVP) paradigm, sound affects participants’ recognition of targets. Although many studies have shown that sound improves cross-modal processing, researchers have not yet explored the effects of sound semantic information with respect to different locations and processing modalities after removing sound saliency. In this study, the RSVP paradigm was used to investigate the difference between attention under conditions of consistent and inconsistent semantics with the target (Experiment 1), as well as the difference between top-down (Experiment 2) and bottom-up processing (Experiment 3) for sounds with consistent semantics with target 2 (T2) at different sequence locations after removing sound saliency. The results showed that cross-modal processing significantly improved attentional blink (AB). The early or lagged appearance of sounds consistent with T2 did not affect participants’ judgments in the exogenous attentional modality. However, visual target judgments were improved with endogenous attention. The sequential location of sounds consistent with T2 influenced the judgment of auditory and visual congruency. The results illustrate the effects of sound semantic information in different locations and processing modalities.

## 1. Introduction

In their daily lives, people often encounter events that are presented to them in a rapid manner, and they therefore overlook some of the information presented in fast sequences. This phenomenon is called attentional blink (AB) [1]; in other words, when multiple stimuli are presented consecutively at the same location, participants’ recognition accuracy for the second target is significantly reduced if there is a 200–500 ms time interval between two targets (target 1, T1; target 2, T2) and if other stimuli are present during that interval. The rapid serial visual presentation (RSVP) stream is a paradigm for the study of attention in the temporal dimension, and given limited cognitive capacity, AB is the standard response [2,3]. As the capacity of the attentional system is limited, the cost of selective attention to a visual stimulus can cause functional blindness to other unattended stimuli [4].

The inhibition model [5] suggests that attentional resources are allocated to the target stimulus after it enters the sensory memory unimodality; a distracting stimulus following the target stimulus and a short underestimation period occur at this time to inhibit the processing of the probe stimulus. These processes are unimodally influenced by lag factors and gradually slow down as the lag increases in the underestimation period. In the same way, AB effects are influenced by perceptual processing. Within a single channel, the interference model [6], which delineates the processing of stimuli in more detail, has been proposed. This model considers the first stage to be the perceptual encoding stage and the second stage to be the central processing stage, which includes short-term memory consolidation, mental rotation, and response selection. Perceptual encoding is performed in parallel, while short-term memory consolidation and response selection take up central resources and thus cannot be performed simultaneously. The phenomenon of attentional transience is caused by the limited capacity of central resources, which are occupied by the short-term consolidation process of the target stimulus; the detection stimulus being perceptually encoded cannot be processed centrally and decays rapidly under interference from the disturbing stimulus.

Cross-modal research is closer to people’s daily lives. Visual and auditory channels interact in daily tasks. How auditory information can better help people recognize visual tasks and how visual tasks can help people improve auditory information are important research questions. Previous studies have shown that the presence of simultaneous sounds with T2 affects the recognition of T2: that is, it improves T2 accuracy [7]. These sounds are mostly nonsense pure tones [7,8] and are accompanied by equal amounts of semantic pure tones [9,10]. Either sound improves the accuracy of T2 recognition, a phenomenon known as the audiovisual enhancement effect [11,12,13]. However, these studies only considered the effect of a single salient stimulus: that is, the sound is a salient stimulus that can significantly attract the attention of others [14]. This is similar to the well-known “cocktail party scenario” [15,16].

Sound features are an important factor influencing cross-modal activation in selective attention [17]. Systematic interactions between participants during visual and auditory cross-modal processing were reported in a study by Evans and Treisman [18,19]. For example, the frequency of the sound affects how the visual space is perceived [20]. A study by Aijun Wang et al. [8] showed that cross-modal attentional enhancement was not modulated by stimulus salience. In their experiments distinguishing between three sound conditions (sound: absent vs. synchronized with T2 vs. synchronized with all), there was no significant difference in behavioral outcomes between synchronization with T2 and synchronization with all, and a non-traditional RSVP paradigm (containing masking) with semantic-free pure tones was used in the study. This was somewhat controversial, given the study of Soh et al. [21], which found that unexpected sounds suppressed representations of visual stimuli. Evans [19] found that cross-modal processing is influenced by the individual perceptual level. The perceptual load similarly affects subjects’ recognition of identifying information. Therefore, if all distractors are accompanied by semantic sounds, increasing the perceptual load may further affect subjects’ accuracy for T2 recognition. Contradicting this, some researchers have argued that attention can be divided into two parts, one for visual and one for auditory tasks, with no interference between the two tasks; furthermore, visual and auditory processing are parallel and independent of each other, and there is specificity in the sensory channels for attentional resources [22].

Related studies have noted that auditory modalities can improve visual attention [23] and have suggested that this may be a form of alertness. It has also been suggested that T2 accuracy is influenced by sound semantics rather than vigilance [9,10]. The RSVP stream is a paradigm that investigates attention in the temporal dimension, and lag has an important influence on the generation of AB [24]. In cross-modal tasks, the serial processing of auditory modalities may affect subjects’ recognition [25]. Therefore, if both auditory and visual sequences are present and change the sequence position of the auditory modalities and T2, the recognition of AB and visual T2 may be affected.

The processing modality can also affect cross-modal integration. One study compared the effects of different processing modalities on cross-modal processing by distinguishing between high and low pitches [8]. The recognition of T2 was influenced by having participants actively process sounds accompanying the RSVP stream, and it was found that top-down processing contributed to improved T2 recognition and improved AB [8]. Cross-modal processing was reported to be nonautomatic in other related studies [26,27]. However, some studies have pointed to the existence of automatic cross-modal processing [18,28,29,30,31], and Kawashima et al. noted the bottom-up auditory entrainment of different frequencies and phases affecting AB [32]. Cross-modal processing is generally considered an automatic process, but it may be influenced by top-down factors. Therefore, distinguishing between processing modalities is essential for cross-modal processing and integration.

To further exclude the influence of sound salience on cross-modal AB and further refine the modulation by which the presence of semantically related sounds affects cross-modal AB, we performed three experiments to distinguish the effects of semantic coherence, the location of sound sequences consistent with T2, and different processing modalities on cross-modal AB. First, we established sounds synchronized with all stimuli to exclude the effect of sound salience on the experiment, and based on that, we distinguished between semantic congruence and incongruence and T2 recognition in the absence-of-sound condition. Second, we further distinguished the effects of early versus lagged sound appearance under semantic and T2 congruence. Finally, a discrimination task was used to distinguish participants’ attentional patterns. Participants were required to discriminate the consistency of sounds synchronized with T2. This ideally distinguished the impact of the task on cross-modal processing across attentional modalities. In Experiment 1, we used the RSVP paradigm, manipulating the characteristics of the presented stimulus modality (no sound, all synchronization congruent with T2, or synchronization incongruent with T2) to test whether semantic congruence and sound characteristics influence cross-modality and whether T2 accuracy is semantically influenced under conditions that exclude high sound salience. In Experiment 2, the sequence position of semantically consistent sounds with T2 was changed. The goal was to examine whether T2 accuracy is affected if semantics appear earlier or later. Experiment 3 built on this by further differentiating the processing modality. Splitting attention into different modalities affects cross-modal processing [33,34,35,36], and we hypothesized that the sequence position of the semantics and the processing mode would affect cross-modal processing and lead to a decrease in T2 accuracy.

## 2. Experiment 1

To investigate whether the semantic-laden sounds accompanying the full stimulus would affect AB, we used capital letters with their pronunciation as the target and sound content in the RSVP task.

### 2.1. Method

#### 2.1.1. Participants

G*Power 3.1.9.2 [37] was used to estimate the sample size for a 3 × 3 two-way repeated-measures analysis of variance (ANOVA) (estimated effect size f = 0.2, alpha = 0.05, power = 0.9), with a suggested sample of 28 participants. Subjects were recruited through posters posted on Soochow University’s online platform and in the surrounding community. In total, 38 subjects participated in Experiment 1. This experiment focused on the effect of speech on AB across audiovisual modalities, so three participants who did not exhibit AB (the mean accuracy of T2T1 in lag3 was greater than the mean accuracy of T1 in the no-sound modality) were excluded; in other words, non-blinkers were excluded [38].

The final 35 subjects were a valid participant population (23 females). Their ages ranged from 19 to 24 years (M = 20.63, standard deviation (SD) = 1.68), and all participants were right-handed and had normal or corrected-to-normal vision. All participants’ native language was Chinese, and they had completed nine or more years of English language learning. All participants had passed China’s College English Test 4. This ensures that they have the ability to dictate in English. All participants were able to accurately identify the listening and visual materials involved in the experiment. None of the participants had a history of neurological or psychiatric disorders, and they all provided written informed consent prior to participation. At the end of the experiment, all participants were paid RMB 30 after completing the experiment, and all study procedures were conducted in accordance with the Declaration of Helsinki. This study was conducted after receiving approval from the ethics committee of Soochow University.

#### 2.1.2. Apparatus, Stimuli, and Experimental Setup

In Experiment 1, all of the visual stimuli were presented on a 19″ screen (DELL-E2316HF) with a gray background (RGB: 196, 196, 196), a resolution of 1280 × 1024, and a refresh rate of 60 Hz. Participants were asked to keep their eyes about 80 cm from the center of the screen during the experiment. All experiments were programmed using MATLAB 2019a and Psychtoolbox 3.1.2 [39], and the data were collated in MATLAB 2019a.

All of the visual stimuli in the experimental material were composed of numbers and letters [40], where the numbers from 0 to 9 constituted distractors, presented in sequences with alternating odd and even numbers to avoid presenting the same numbers consecutively. Capital letters were used as the first target (T1) and second target (T2). Non-monosyllabic letters were excluded, and confusing letters were excluded. The final letters E, G, J, U, V, R, and B were used for T1, and K, T, P, A, Z, N, and Y were used for T2. Subjects were informed about the letters that would appear before they performed the experiment, but not about which letters were used as targets; that is, they were not informed about the specifics of T1 and T2.

The auditory material was created using Adobe Audition 3.0 sound production software, and the stimuli were pure tones (1259 Hz) generated by a 75 dB sine wave with a duration of 75–80 ms (different-sounding letters did not necessarily play for the same duration, but were within that range). The stimulating pronunciation was the voice of a middle-aged woman to avoid the influence of tone and other factors [41,42]. All auditory stimuli were started at the same time as the visual stimuli on the same screen. The sound was played through wired headphones (BOSE QuietComfort QC20), and all participants reported hearing the same sound content in both ears, without a delay, at the same time as the visual stimuli. We had all subjects individually listen to the accompanying audio files (all letter sounds selected in our experiment) separately before the start of Experiment 1. All participants accurately reported the letter corresponding to the audio file.

#### 2.1.3. Experimental Procedures and Design

This experiment used a 3 (sound: no sound vs. consistent with T2 vs. inconsistent with T2) × 3 (lag: lag1 vs. lag3 vs. lag8) within-subject design. In total, there were 9 conditions, 48 trials per condition, and 432 trials. The experimental procedure is shown in Figure 1. At the beginning of each trial, a fixation cross was presented in the center of the screen for 1000 ms, and all subjects were asked to look at it. An RSVP stimulus stream consisting of 23 elements was then presented, with 2 targets (letter targets) and 21 number distractors (non-targets) in each stream. After each stimulus had been presented for 100 ms (visual stimulus for 87 ms and blank screen for 17 ms), the next stimulus was presented immediately. In other words, the stimulus onset asynchrony (SOA) was only 100 ms. An SOA between T1 and T2 of 100 ms indicated lag1; lag3 indicated an SOA of 300 ms for T1 and T2, and lag8 indicated an SOA of 800 ms for T1 and T2. After all of the stimuli for each RSVP stimulus stream had been presented, a screen without any stimuli was presented for 100 ms. After that, two questions (1: What was the first letter you saw? 2: What was the second letter you saw?) appeared in the center of the screen, and the second question was displayed only after the subject had finished answering the first one. All subjects were asked to respond to the questions by making corresponding keystroke responses on the keyboard, and the process only recorded the accuracy without requiring subjects to respond quickly and without recording response times.

Except for the no-sound condition, all stimulus conditions were accompanied by a sound with a different meaning than the stimulus. Moreover, all sounds appeared at the same time as the stimulus on the screen, the sound was played for 75 ms before it was stopped, and a different sound file was played when the next stimulus appeared.

During the experiment, subjects were asked not to focus on the sound. At the end of the experiment, each participant was asked about the position of the letter and number distractors in relation to the sound on the same screen, and they all reported that they had appeared at the same time. Each subject performed 36 practice sessions before the formal experiment, which began after the subjects understood the experiment.

#### 2.1.4. Statistical Analysis

The profile analysis had two dependent variables: the correct T1 recognition rate (the rate of correctly judging the first target) and the correct T2T1 recognition rate (the rate of correctly judging the second target of the first target). The results were tested using the Shapiro–Wilk (S-W) test (*p* > 0.05). All data satisfied a normal or approximately normal distribution. Referring to the study by Liu et al. [43], we excluded all data beyond three standard deviations. After calculating T1 and T2T1 accuracies, further analysis was performed. The data were analyzed with SPSS version 24.0 for Windows. Continuous variables with a normal distribution were presented as the mean ± standard deviation (SD). A value of *p* < 0.05 was considered significant for all experiments.

### 2.2. Results

First, one-way ANOVA was conducted for the three sound conditions (no sound, consistent sound with T2, and inconsistent sound with T2), with the correct T1 recognition rate used as the dependent variable to analyze whether the main effect of the sound condition was significant and to compare the differences among the three conditions. Next, a two-way repeated-measures ANOVA with a 3 (sound: no sound vs. consistent with T2 vs. inconsistent with T2) × 3 (lag: lag1 vs. lag3 vs. lag8) design was conducted using the rate of correct T2T1 recognition as the dependent variable. The main effects of the sound conditions, lag conditions, and their interaction were analyzed for significance, followed by a simple effects analysis to compare the differences between sound conditions.

The overall correct T1 recognition rate was 93.6 ± 10.3% (kurtosis −2.89, skewness 12.7), and a one-way ANOVA with T1 as the dependent variable was performed. The results showed a significant main effect for sound (*F*(2, 313) = 13.293, *p* < 0.001, *η_P_*^2^ = 0.131). A post hoc test showed that the no-sound (89.01 ± 12.2%) condition was significantly different from the consistent sound with T2 condition (95.6 ± 5.7%) vs. the inconsistent sound with T2 condition (96.35 ± 8.5%) (*t*(105) = 3.94, *p* < 0.001, *Cohen’s D* = 0.73, 95% CI = [0.033, 0.099]; *t*(105) = 4.26, *p* < 0.001, *Cohen’s D* = 0.34, 95% CI = [0.039, 0.107], respectively). Further, there was no significant difference between the consistent sound with T2 and inconsistent sound with T2 conditions (*p* > 0.05) (Table 1).

The accuracy of T1 judgments was influenced by the presence of concomitant sounds. The overall correct T2T1 recognition rate was 88.6 ± 14.5% (kurtosis −1.67, skewness 2.9), and a two-way repeated-measures ANOVA with T2T1 as the dependent variable was conducted. The results showed that the main effect of sound was significant (*F*(2,105) = 15.082, *p* < 0.001, *η_P_*^2^ = 0.146), indicating that sound significantly affected subjects’ accuracy for T2. The main effect of the lag condition was significant (*F* (2,105) = 7.522, *p* = 0.001, *η_P_*^2^ = 0.078), indicating that the difference between the T2 and T1 lag conditions affected subjects’ accuracy for T2. The interaction between voice and lag was not significant (*p* > 0.05), indicating that these two factors were independent of each other in their influence on the accuracy of subjects’ judgments of T2. The values were corrected using the Bonferroni method (*p* < 0.05) (Table 1).

In a further simple effects analysis, we found a significant sound effect when T2 was at lag1 (*F*(2,70) = 6.391, *p* = 0.003, *η_P_*^2^ = 0.176). The no-sound condition (80.36 ± 13.2%) was significantly different from the consistent sound with T2 (91.96 ± 9.6%) vs. the inconsistent sound with T2 conditions (92.86 ± 12.6%) (*t* [33] = 2.895 *p* = 0.006 *Cohen’s D* = 0.99, 95% CI = [0.035, 0.197]; *t*(35) = 3.21 *p* = 0.003 *Cohen’s D* = 0.89, 95% CI = [0.046, 0.203], respectively). The latter two conditions were not significantly different (*p* > 0.05). This indicates that in the lag1 condition, the voice synchronized with the sequence affected the accuracy of the subjects’ judgment of T2. However, there was no significant effect of sounds that were semantically consistent and inconsistent with T2 on the experimental effects. The values were corrected using the Bonferroni method (*p* < 0.05) (Table 1).

We found that the sound effect was significant when T2 was at lag3 (*F*(2,70) = 7.489, *p* = 0.001, *η_P_*^2^ = 0.2). The no-sound (73.81 ± 19.6%) condition was significantly different from the consistent sound with T2 (88.99 ± 8.6%) vs. the inconsistent sound with T2 conditions (90.77 ± 13.3%) (*t*(35) = 2.99, *p* = 0.005, *Cohen’s D* = 0.92, 95% CI = [0.049, 0.254]; *t*(35) = 3.28, *p* = 0.002, *Cohen’s D* = 1.01, 95% CI = [0.065, 0.273], respectively). The latter two conditions were not significantly different (*p* > 0.05), which indicates that, in the lag3 condition, the voice synchronized with the sequence affected the accuracy of the subjects’ judgment of T2. However, there were no significant effects for sounds that were semantically consistent and inconsistent with T2 on the experimental effects. The values were corrected using the Bonferroni method (*p* < 0.05) (Table 1).

We found that the sound effect was not significant when T2 was at lag8. Significant differences were found only in the no-sound (88.99 ± 14.9%) condition versus the consistent with T2 condition (96.73 ± 7.6%) (*t*(35) = 2.06, *p* = 0.045, *Cohen’s D* = 0.63729, 95% CI = [0.002, 0.153]). This indicates that T2 recognition at lag8 could be significantly improved under the semantic consistency condition (Table 1).

The analysis of the results of Experiment 1 showed that semantically irrelevant sounds synchronized with visual stimuli significantly reduced AB regardless of whether they were consistent with T2: that is, cross-modal processing contributes to attention (Figure 2). Previous studies have also shown that attention similarly modulates cross-modal processing [38].

How can semantically rich information, even in real life, influence attention through changes in semantics? Research has shown that information that is relevant to the organism’s current goal (i.e., relevant to the task) is prioritized for attention [44,45]. Similarly, insignificant and task-irrelevant information also affects attentional priority [44]. Therefore, in Experiment 2, we altered the auditory modality stimuli and removed the no-sound condition by playing the auditory sounds that were semantically congruent with T2 earlier or later to explore whether the advancement or lagging of sounds congruent with T2 could affect the allocation of attention—that is, the rate of accuracy for T2.

## 3. Experiment 2

Experiment 1 demonstrated that a sound containing semantics improved the accuracy in recognizing T1 vs. T2T1 and improved the AB effect. However, there was no clear conclusive evidence of whether sounds containing semantics interfered with subjects’ perceptions. Study 2 thus omitted two sound conditions (no sound and inconsistent with T2) and added two sound conditions (sound consistent with T2 played before lag2 vs. sound consistent with T2 played after lag2). We thus investigated whether the sound’s advanced or lagged playback associated with T2 would affect the recognition of T2. If the inhibition model is applied to cross-channel tasks, then semantic sounds would affect T2 recognition because it would reinforce the perceptual load of T1. Conversely, if visual processing and auditory processing are simultaneous, then the interference model would be confirmed. We hypothesized that the early appearance of a sound consistent with T2 would interfere with T2 recognition and reduce the correct T2 recognition rate.

### 3.1. Method

#### 3.1.1. Participants

The final 32 subjects were a valid participant population (17 females). Their ages ranged from 19 to 24 years (M = 20.84, SD = 2.01). All participants were right-handed and had normal or corrected-to-normal vision. None of the participants had a history of neurological or psychiatric disorders, and they all provided written informed consent prior to participation. At the end of the experiment, all participants were paid, and the study guidelines were in accordance with the Declaration of Helsinki.

#### 3.1.2. Apparatus, Stimuli, and Experimental Setup

In Experiment 2, the condition of sound consistent with T2 was retained, and two sound conditions, consistent sound with T2 played 200 ms earlier (defined as T2 sound earlier) and consistent sound with T2 played 200 ms later (defined as T2 sound later), were added. The auditory stimuli and all other stimuli and devices were the same as in Experiment 1.

#### 3.1.3. Experimental Procedures and Design

The experimental procedure for Experiment 2 was the same as that for Experiment 1; however, two independent variables were changed in Experiment 2: that is, the no-sound and inconsistent T2 sound conditions were removed, and the T2 sound earlier and T2 sound later conditions were added. Thus, Experiment 2 was a 3 (sound: consistent with T2 vs. T2 sound earlier vs. T2 sound later) × 3 (lag: lag1 vs. lag3 vs. lag8) within-subject design. In total, there were 9 conditions with 48 trials each, for a total of 432 trials. Before the start of the experiment, participants performed 24 practice trials to familiarize themselves with the experiment.

#### 3.1.4. Statistical Analysis

The statistical analysis was the same as for Experiment 1. The profile analysis had two dependent variables: the correct T1 recognition rate and the correct T2T1 recognition rate. The results were tested using the S-W test (*p* > 0.05). All data satisfied a normal or approximately normal distribution. Referring to the study by Liu et al. [38], we excluded all data beyond three standard deviations. After calculating the T1 and T2T1 accuracies, further analysis was performed.

### 3.2. Results

First, a one-way ANOVA was conducted for the three conditions of sound (consistent sound with T2 vs. T2 sound earlier vs. T2 sound later), with the correct T1 recognition rate as the dependent variable to analyze whether the main effect of the sound condition was significant and to compare the differences among the three conditions. Next, a two-way repeated-measures ANOVA for a 3 (sound: consistent with T2 vs. T2 sound earlier vs. T2 sound later) × 3 (lag: lag1 vs. lag3 vs. lag8) design was conducted using the rate of correct T2T1 recognition as the dependent variable. The main effects of sound conditions, lag conditions, and the interaction between them were analyzed for significance, followed by a simple effects analysis to compare the differences between sound conditions.

The overall correct T1 recognition rate was 91.43 ± 14.3% (kurtosis −3.35, skewness 13.6), and a one-way ANOVA with T1 as the dependent variable was performed. The results showed that the lag main effect was not significant (*F* = 2.221, *p* > 0.05). There was no significant difference between the lag1 (89 ± 16.6%) and lag3 (91.4 ± 15.6%) conditions according to the post hoc test (*p* > 0.05), and there was no significant difference between the lag3 and lag8 (93.8 ± 9.3%) conditions (*p* > 0.05). However, there was a significant difference between lag1 and lag8 (*t*(2,154) = −2.224, *p* < 0.05, *Cohen’s D* = −0.36, 95% CI = [−0.091, −0.005]). The accuracy of the T1 judgment appeared to be affected by the lag condition (Table 2).

The overall correct T2T1 recognition rate was 86.53 ± 18% (kurtosis −2.45, skewness 6.9), and two-way repeated-measures ANOVA was performed. The results showed that the main effect for sound was not significant (*F* = 0.029, *p* > 0.05). The lag main effect was not significant (*F* = 1.506, *p* > 0.05), and the sound–lag interaction was also not significant (*F* = 0.221, *p* > 0.05) (Table 2). This indicated that overall, in the RSVP stream, a sound consistent with T2 played in advance, simultaneously, or with a lag hardly affected the recognition accuracy for T2 at different lag positions (Figure 3).

After the simple effects analysis, the sound condition was not significant in the lag1 condition (*F* = 0.255, *p* > 0.05), in the lag3 condition (*F* = 0.313, *p* > 0.05), or in the lag8 condition (*F* = 0.183, *p* > 0.05). In summary, the results for Experiment 2 showed that playing the speech before appearance of T2, playing it simultaneously with T2, and playing it after the appearance of T2 did not affect the accuracy of subjects’ recognition of T2. These results are different from those of previous studies (Table 2).

Some studies have suggested that subjects’ perceptual systems are sensitive to semantically consistent sounds [46]. A study by Song Zhao et al. [9] showed that the presence of T2-consistent vocal information alone activated participants’ recognition of T2 significantly more than inconsistent information. We consider that there are two main reasons for the absence of significant effects for advanced versus lagged semantic information: First, the sound accompanying all stimuli reduced the sound salience of the T2 stimulus sound. Second, probably due to the influence of the processing style, for bottom-up information, subjects may ignore some semantic information.

In Experiment 3, we prompted the top-down processing of sound information by adding a new question in which subjects were asked to determine whether the semantic meaning of the sound that appeared simultaneously with T2 was consistent with the T2 letters. This question was used to further clarify whether the early or late appearance of semantic information under top-down versus bottom-up processing affected T2 accuracy.

## 4. Experiment 3

Experiment 2 built on Experiment 1 and demonstrated that semantics also affected the correctness of T1 versus T2T1. It is not known whether this effect is influenced by the processing style, and there may be significant differences between bottom-up and top-down processing. Experiment 3 added a top-down processing task on top of the bottom-up task. We assumed that the top-down approach would yield greater accuracy than the bottom-up approach.

### 4.1. Method

#### 4.1.1. Participants

The final 35 subjects were a valid participant population (21 females). Their ages ranged from 19 to 24 years (M = 20.67, SD = 1.83). All participants were right-handed and had normal or corrected-to-normal vision. None of the participants had a history of neurological or psychiatric disorders, and they all provided written informed consent prior to participation. At the end of the experiment, all participants were paid, and the study was conducted in accordance with the Declaration of Helsinki.

#### 4.1.2. Apparatus, Stimuli, and Experimental Setup

In Experiment 3, all conditions of Experiment 2 were retained, and the auditory stimuli and all other stimuli and devices were matched to those of Experiment 1.

#### 4.1.3. Experimental Procedures and Design

The experimental procedure for Experiment 3 was the same as that for Experiment 2; a 3 (consistent sound with T2 vs. T2 sound earlier vs. T2 sound later) × 3 (lag: lag1 vs. lag3 vs. lag8) within-subject design was again used. In total, there were 9 conditions with 48 trials each, for a total of 432 trials. After the entire stream of RSVP stimuli appeared, subjects were asked to answer three questions that appeared sequentially in the center of the screen (1: What was the first letter you saw? 2: What was the second letter you saw? 3: Was the second letter the same as the one you heard?); each question was displayed only after the subject had completed the previous question. Before the start of the experiment, participants performed 24 practice trials to familiarize themselves with the experiment.

#### 4.1.4. Statistical Analysis

The profile analysis had three dependent variables: the correct T1 recognition rate, correct T2T1 recognition rate, and correct T2VA recognition rate. The results were tested using the S-W test (*p* > 005). All data satisfied a normal or approximately normal distribution. Referring to the study by Liu et al. [38], we excluded all data beyond three standard deviations. After calculating T1, T2T1, and T2VA accuracies, further analysis was performed.

### 4.2. Results

One-way ANOVA was conducted for the three lag conditions (lag1 vs. lag3 vs. lag8) using the rate of correct T1 recognition as the dependent variable. The main effects of lag conditions were analyzed for significance, and the differences among them were compared. A two-way repeated measures ANOVA of a 3 (consistent sound with T2 vs. T2 sound earlier vs. T2 sound later) × 3 (lag1 vs. lag3 vs. lag8) design was conducted using the rate of correct T2T1 recognition as the dependent variable. The main effects of sound conditions, lag conditions, and their interaction were analyzed for significance, followed by a simple effects analysis to compare the differences between the sound conditions. A two-way repeated-measures ANOVA with a 3 (sound: consistent sound with T2 vs. T2 sound earlier vs. T2 sound later) × 3 (lag: lag1 vs. lag3 vs. lag8) design was also conducted using the rate of correct T2VA consistency judgments as the dependent variable. The main effects of sound conditions, lag conditions, and their interaction were analyzed for significance, followed by a simple effects analysis to compare the differences between sound conditions.

The overall correct T1 recognition rate was 94.93 ± 6.2% (kurtosis −1.65, skewness 2.8), and a one-way ANOVA with T1 as the dependent variable was performed. The results showed that the lag main effect was not significant (*F* = 0.472, *p* > 0.05). This indicates that, overall, in the RSVP stream, the accuracy of T1 was not affected by the lag factor (Table 3).

The overall correct T2T1 recognition rate was 91.13 ± 11.3% (kurtosis −1.99, skewness 4.2), and two-way repeated-measures ANOVA was performed. The results showed that the sound main effect was not significant (*F* = 0.081, *p* > 0.05); the lag main effect was not significant (*F* = 2.289, *p* > 0.05); and the sound–lag interaction was not significant (*F* = 0.202, *p* > 0.05) (Table 3). This shows that, overall, in the RSVP stream, playing sounds consistent with T2 earlier, simultaneously, or later did not significantly affect the recognition accuracy of T2 at different lag locations (Figure 4).

Comparing the differences between the lag conditions for all sound conditions, there was a significant difference between lag1 (92.61 ± 8%) and lag8 (89.3 ± 13.8%) (*t*(158) = 1.994, *p* = 0.048, *Cohen’s D* = 0.27, 95% CI = [0.004, 0.066]). The lag condition was not significant in the consistent with T2 condition (*F* = 1.724, *p* > 0.05). In the T2 sound earlier condition, the lag condition was not significant (*F* = 0.36, *p* > 0.05). After a pairwise comparison, there was no significant difference between lag8 (88.5 ± 15.1%) and lag1 (93.4 ± 8.6%) (*t*(53.77) = 1.64, *p* = 0.072, *Cohen’s D* = 0.393, 95% CI = [−0.011, 0.109]) (Table 3).

T1 and T2T1 in Experiment 3 and Experiment 2 were used as dependent variables to compare the differences in accuracy using different processing methods. The results showed that the main effect of the processing mode was significant (*F* = 13.3, *p* < 0.05). After further simple effects analysis, there was no significant difference in the correctness of T1 in the conditions of sound consistent with T2, T2 sound earlier, and T2 sound later regarding the processing mode (*p* > 0.05). In lag1, the T2 sound later condition was significantly affected by the processing method (*t*(32.6) = 2.33, *p* = 0.026, *Cohen’s D* = 1.121, 95% CI = [0.098, 0.144]) (Figure 5). There was no significant difference in sound consistent with T2 and T2 sound earlier conditions (*p* > 0.05) (Figure 6). In lag3, the consistent with T2 condition was significantly affected by the processing method (*t*(60) = 2.09, *p* = 0.041, *Cohen’s D* = 0.456, 95% CI = [0.036, 0.148]) (Figure 7). There was no significant difference in the T2 sound later and the T2 sound earlier conditions (*p* > 0.05). There was no significant difference in the lag8 condition in the sound consistent with T2, T2 sound later, and T2 sound earlier conditions (*p* > 0.05).

The overall correct T2VA recognition rate was 66.82 ± 23.2% (kurtosis −0.68, skewness −0.37), and two-way repeated-measures ANOVA was performed. The results showed a significant voice main effect (*F*(2, 314) = 35.83, *p* < 0.001, *η_P_*^2^ = 0.19). The lag main effect was significant (*F*(2, 314) = 4.95, *p* = 0.008, *η_P_*^2^ = 0.031). Sound and lag interacted significantly (*F*(2, 314) = 3.054, *p* = 0.017, *η_P_*^2^ = 0.038). This showed that in judging T2VA consistency, sound conditions, lag conditions, and their interactions could affect the judgment of T2 consistency.

After a simple effects analysis, the sound condition was significant in the lag1 condition (*F*(2,102) = 14.36, *p* < 0.001, *η_P_*^2^ = 0.22). Significant differences were found in the consistent with T2 (62.42 ± 22.5%) and T2 sound later (76.16 ± 16.6%) conditions (*t*(68) = 2.83, *p* = 0.006, *Cohen’s D* = 0.68, 95% CI = [0.041, 0.234]). Significant differences were found in the consistent with T2 and T2 sound earlier (48.95 ± 22.8%) conditions (*t*(68) = 2.436, *p* = 0.017, *Cohen’s D* = 0.27, 95% CI = [−0.245, −0.024]). Significant differences were found in the T2 sound earlier and T2 sound later conditions (*t*(68) = −5.65, *p* < 0.001, *Cohen’s D* = −1.35, 95% CI = [−0.368, −0.176]). The values were corrected using the Bonferroni method (*p* < 0.05) (Figure 8).

Sound conditions were also significant under lag3 conditions (*F*(2,102) = 22.45, *p* < 0.001, *η_P_*^2^ = 0.31); significant differences were found in the consistent with T2 (56.5 ± 20.1%) and T2 sound later (83.4 ± 12.9%) conditions (*t*(68) = 6.72, *p* < 0.001, *Cohen’s D* = 1.61, 95% CI = [0.189, 0.348]). There was no significant difference between the consistent with T2 and T2 sound earlier (60.85 ± 19.8%) conditions (*t*(68) = −0.74, *p* > 0.05). Significant differences were found in the T2 sound earlier and T2 sound later conditions (*t*(68) = −5.51, *p* < 0.001, *Cohen’s D* = −0.87, 95% CI = [−0.307, −0.134]). The values were corrected using the Bonferroni method (*p* < 0.05) (Figure 8).

There were sound conditions under the lag8 condition (*F*(2,102) = 5.552, *p* = 0.005, *η_P_*^2^ = 0.098); significant differences were found in the consistent with T2 (68.14 ± 25.3%) and T2 sound later (81.29 ± 20.5%) conditions (*t*(68) = 2.38, *p* = 0.02, *Cohen’s D* = 0.57, 95% CI = [0.021, 0.241]). There was no significant difference between the consistent with T2 and T2 sound earlier (63.92 ± 22.9%) conditions (*t*(68) = −0.74, *p* > 0.05). Marginally significant differences existed between T2 sound earlier and T2 sound later conditions (*t*(68) = −3.38, *p* = 0.001, *Cohen’s D* = −0.81, 95% CI = [−0.276, −0.071]). The values were corrected using the Bonferroni method (*p* < 0.05) (Figure 8).

## 5. General Discussion

We examined the effects of cross-modal processing on attentional allocation in three experiments. By comparing semantic information, different temporal locations of semantically congruent information, and differences between processing modalities, we further clarified whether meso-auditory information in cross-modal processing affects accuracy in a visual channel task. Experiment 1 showed that in the no-sound condition, the paradigm was similar to that in previous studies, showing the AB phenomenon in lag3 [47]. The same experiment showed the effect of semantically consistent versus inconsistent auditory information on T2 accuracy in the RSVP stream after the elimination of sound salience in Experiment 1. This is similar to the results of previous studies [8,48]. We excluded the effect of the high salience of individual auditory stimuli on the T2 task in the RSVP stream. All visual stimuli in this model were accompanied by an auditory sound containing semantic meaning. These data suggest that the recognition of T2 targets facilitates subjects’ recognition of T2 targets regardless of whether they contain consistent semantic information. This extends the findings of Wang et al. [47].

Experiment 1 verified that perceptual encoding processing occurred in parallel. Participants could perceive sound and visual stimuli simultaneously, which contributed to the improvement in AB as well as that in the second stage. This indicates that the perceptions of the two channels might influence each other. An improvement in T2 correctness with an increasing stimulus size across T1 channels was demonstrated in Experiment 1. This result conflicts somewhat with the results of the study by Catherine and Jolicoeur [49], which suggested that increasing the T1 difficulty and T1 task load would affect the size and timing of AB, but this is limited to tasks between single channels. Notably, increasing the perceptual load for T1 in cross-channel experiments did not increase AB.

In Experiment 2, we believed that the early appearance or lagged appearance of semantics would affect the AB size, but the actual experiment did not achieve the expected result. This has some commonalities with the cocktail-party phenomenon [50]; participants block out the semantic meaning of some sounds, and this blocking is influenced by the position of the sequence. The same semantically rich sounds are tied to unrelated targets, and the semantic information of the targets is not actively processed. In the semantic congruency task, some studies have found that in the semantic congruency multisensory condition, the response times for visual and auditory sensory memory encoding were faster than those in the single-sensory memory encoding condition [51]. In contrast, the corresponding conclusions were not successfully drawn in Experiment 2 after eliminating the effect of sound saliency. We believe that the saliency of sound is an important way to influence human perceptual processing. The lack of a significant difference between early or delayed sound playback consistent with the semantics of T2 may be related to the playback duration [52]. In other words, 75 ms of semantically rich sounds were presented in the RSVP stream, and subjects may have had difficulty completing their perceptual processing of the semantics. However, the semantics of other voices were irrelevant for participants after eliminating the effect of voice salience, so participants automatically blocked the vast majority of the semantic information, possibly including the semantic information consistent with T2, as this is a non-sparing way of processing their perception [53,54].

Some studies have suggested that cross-modal processing is automated [8]. Other studies have suggested, however, that certain phenomena, such as auditory-evoked contralateral occipital positivity (ACOP), arise under specific sound conditions only when, for example, participants are asked to distinguish sound locations [17]. Endogenous and exogenous attention are often considered to be two systems in the study of vision [55], and the experimental results in Experiment 2 and Experiment 3 suggest that there may also be two systems of endogenous and exogenous attention in the cross-modal task.

In Experiment 3, a decreasing trend appeared (shown in Figure 3), which is different from the findings of previous studies [8,9,10]. The correct recognition rate at lag8 should be higher than that at lag1 and lag3 (i.e., T2 occurring within 500 ms), as in previous studies of cross-modal RSVP streams. In contrast, Experiment 2 and Experiment 3 showed an insignificant difference at lag8, which is related to the growth of the perceptual load with the lag factor. We believe that this is somewhat related to the processing method.

A further comparison was made between Experiment 2 and Experiment 3, which further confirmed the existence of a major effect of the processing mode. With top-down processing, the sound was more likely to affect the subjects’ accuracy. Processing in a top-down mode can help participants extract more information. However, a significant difference was only shown in the T2 sound later condition. This may be related to the presence of the task paradigm [56]. The auditory information improved the correctness of the RSVP stream, and adding the processing modality and semantic information to it resulted in an insignificant increase in accuracy. The increase in the perceptual load was, to some extent, offset by the semantic advantage of active processing [57]. Notably, these results were only seen in Experiment 3, when participants could potentially ignore the semantic information related to T2.

In addition, subjects in Experiment 3 were required to answer three questions but showed a higher rate of accuracy for T2T1 compared to Experiment 2. Previous studies have posited that the more targets in the RSVP stream, the lower the correct T2T1 recognition rate [58]. However, higher levels of performance were shown during inter-channel tasks. This could be related to the endogenous, target-driven selective modulation of multisensory performance [59]. In other words, endogenously attending to processing reduces the number of times participants ignored irrelevant information. In addition, cognitive resources shared between the visual and auditory channels may have been mobilized during cross-modal task transitions [60]. In other words, more attentional resources were activated in a multichannel task, and the sum of the resources was greater than in a single-channel versus bottom-up processing modality. This is an interesting phenomenon. When faced with events in daily life that need to be presented in a rapid sequence, attention can be improved and AB can be reduced by adding an accompanying sound. This accompanying sound can be information related to the target or it can be inconsistent with the target event. Fully engaging auditory perception further improves the processing efficiency of visual information. We believe that Q3 in Experiment 3 was a task that responds well to audiovisual integration—that is, the correct judgment of T2VA consistency in Experiment 3. This task effectively illustrated that visual and auditory tasks can be processed simultaneously or nearly simultaneously.

The semantic agreement of information with T2 in the T2VA consistency judgment task affected the judgment of T2 sound consistency. This possibly confirms the interference model, in which perceptual processing conflicts with attentional transients, with more attentional resources required at lag3. This appeared in the T2 sound earlier condition as well, which may be related to the presence of T1 enhancement [61], allowing subjects to focus more attentional resources on the visual channel task. The early appearance of sounds increased the difficulty of the T1 task [49], which affected the judgment of auditory information. The early appearance of speech information interfered with the sounds that appeared simultaneously with T2 [59]. The T2 sound later condition showed higher accuracy, which further demonstrated that the early emergence of semantic information interfered with the target consistency judgments in the same channel.

There was a significant difference in the accuracy of T2T1 versus T2VA agreement in Experiment 3. This is similar to a visually highly focused task, which showed similar results in a cross-channel task [62]. We suggest that this reflects an asymmetry in selective attention between cross-channel effects [63]. Auditory stimuli can effectively complement participants’ attention to visual tasks, but the visual system is, to some extent, ineffective in helping participants discriminate sounds in auditory tasks. This is probably related to the dominance of visual perception in daily life.

The present study explored semantic consistency as well as differences in semantic information consistent with T2 across sequence positions and across processing modalities with the effect of sound saliency removed. The paradigm showed AB in a visual-only channel task, but not in a cross-modal task. The sound itself improved the accuracy of T2 recognition.

The present study has some limitations. It used simple monosyllabic letter sounds that conveyed less semantic information, but there are limitations inherent to increasing the perceptual load through monosyllabic letter-sound information. In addition, all of the subjects participated in three experiments on separate days. However, the experimental order of pre-testing according to Experiment 1, Experiment 2, and Experiment 3 may have affected subsequent experiments. The sound presentation time was limited to 75–80 ms, and we compressed the sound quality of the letters read. Although all subjects accurately reported the meaning of all syllables before the experiment, there may still have been negative effects on accuracy owing to sound problems in the RSVP stream. In addition, some semantic information may be present in the timbre and pitch of the voice, such as the voice of a lover. The subjects were all young, and they had better vision, hearing, and reflexes than elderly and other special populations, so the results may not be generalizable.

This study found that auditory tasks are conducive to the recognition of RSVP visual information regardless of semantics, which is helpful for RSVP tasks in people’s daily lives. However, visual information does not necessarily promote the processing of auditory information. In the future, we will further explore how visual and auditory information interfere with each other, and whether other factors in auditory information will affect the recognition of semantic and visual information. Further attempts can be made in future research on how to use speech and semantic information accompanying sounds to better help people in RSVP scenario tasks.

## 6. Conclusions

In this study, cross-modal processing significantly improved AB, but the difference between semantically consistent and inconsistent sound information with T2 did not affect T2 recognition in the visual task after eliminating sound saliency. Moreover, the advanced or lagged appearance of sound consistent with T2 improved target judgments under endogenous attention and did not affect participants’ judgments under the exogenous attentional modality. The early, simultaneous, or lagged appearance of sounds consistent with T2 influenced the judgments of auditory and visual congruence. As such, the results illustrate the effects of auditory semantic information in different temporal locations and processing modalities.

## Figures and Tables

**Figure 1 brainsci-13-00340-f001:**
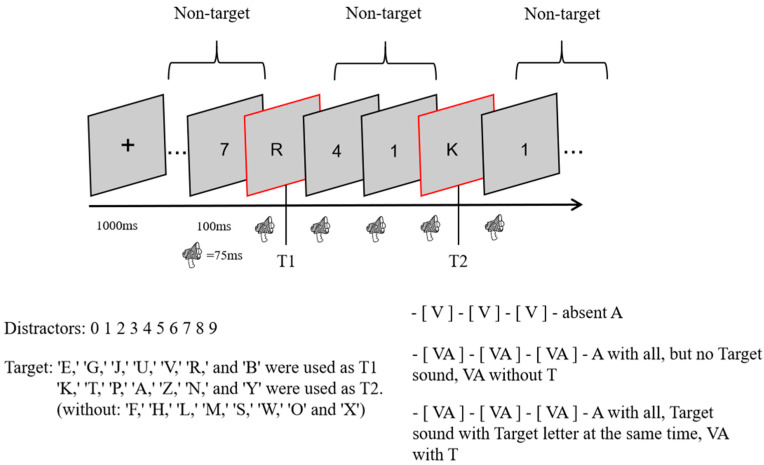
One RSVP stream experiment.

**Figure 2 brainsci-13-00340-f002:**
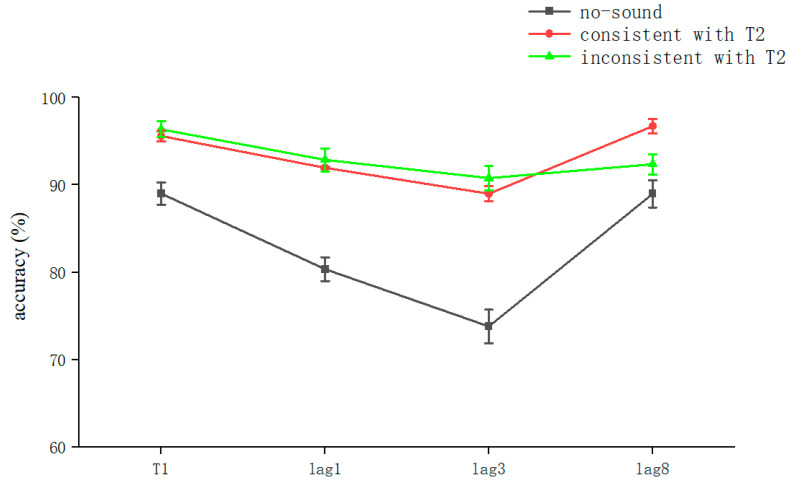
Accuracy shown as a function of the experimental conditions in Experiment 1. Error bars represent the standard error of the mean.

**Figure 3 brainsci-13-00340-f003:**
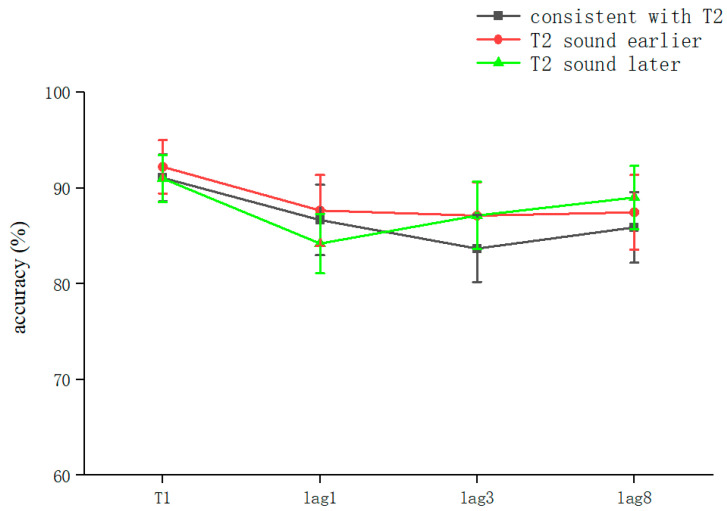
Accuracy shown as a function of the experimental conditions in Experiment 2. Error bars represent the standard error of the mean.

**Figure 4 brainsci-13-00340-f004:**
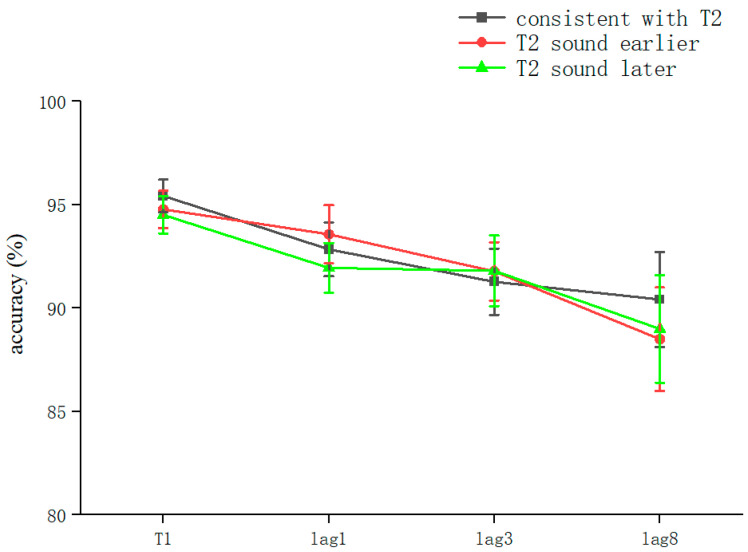
Accuracy shown as a function of the experimental conditions in Experiment 3. Error bars represent the standard error of the mean.

**Figure 5 brainsci-13-00340-f005:**
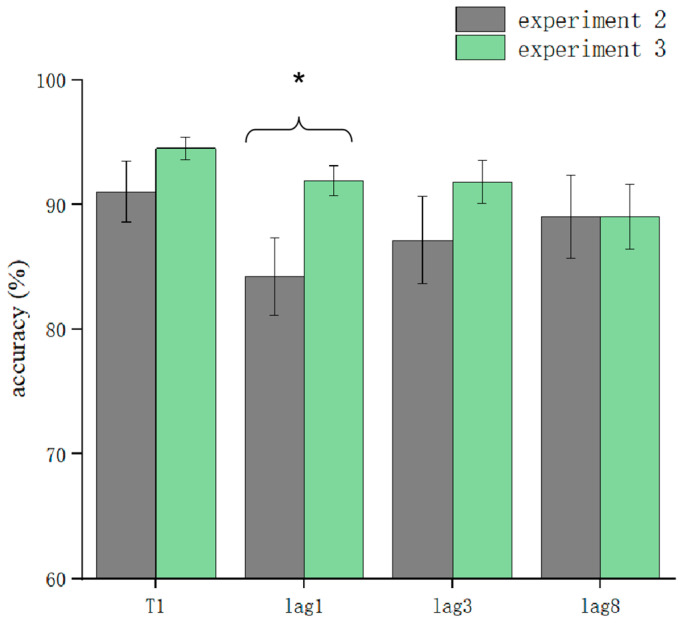
Comparison of accuracy between Experiment 2 and Experiment 3 in the T2 sound later condition (* *p* < 0.05).

**Figure 6 brainsci-13-00340-f006:**
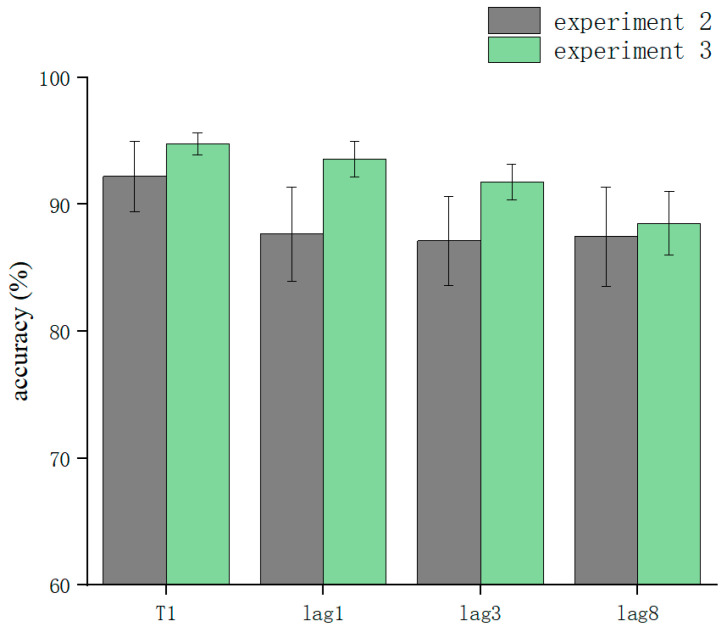
Comparison of accuracy between Experiment 2 and Experiment 3 in the T2 sound earlier condition.

**Figure 7 brainsci-13-00340-f007:**
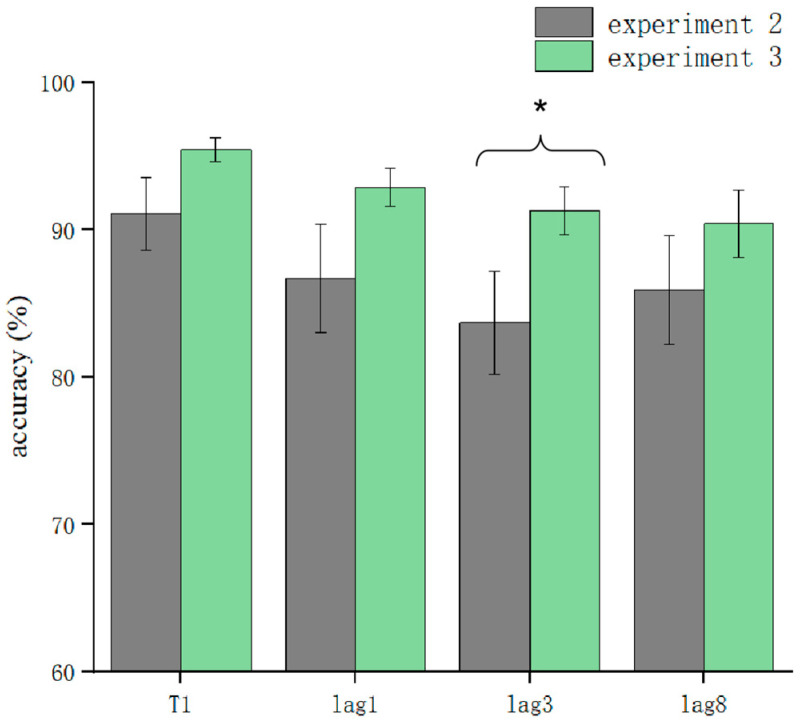
Comparison of accuracy between Experiment 2 and Experiment 3 in the consistent sound with T2 condition (* *p* < 0.05).

**Figure 8 brainsci-13-00340-f008:**
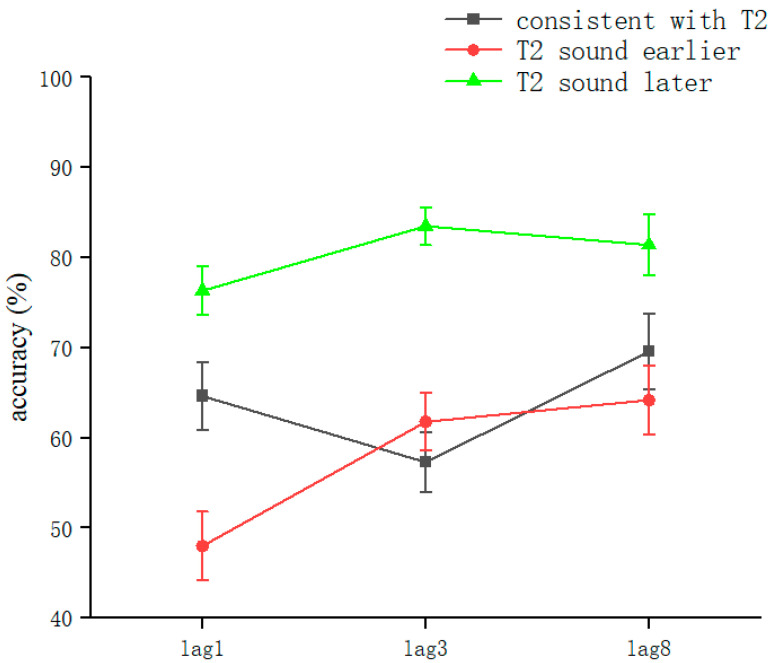
Accuracy of T2VA consistency judgments in Experiment 3. Error bars represent the standard error of the mean.

**Table 1 brainsci-13-00340-t001:** Descriptive statistics of Experiment 1.

	T1 (Mean ± SD) (%)	T2T1 (Mean ± SD) (%)		
		Lag1	Lag3	Lag8
No sound	89.01 ± 12.2	80.36 ± 13.2	73.81 ± 19.6	88.99 ± 14.9
Consistent with T2	95.6 ± 5.7	91.96 ± 9.6	88.99 ± 8.6	96.73 ± 7.6
Inconsistent with T2	96.35 ± 8.5	92.86 ± 12.6	90.77 ± 13.3	92.76 ± 10.8

**Table 2 brainsci-13-00340-t002:** Descriptive statistics of Experiment 2.

	T1 (Mean ± SD) (%)	T2T1 (Mean ± SD) (%)		
		Lag1	Lag3	Lag8
T2 sound earlier	92.1 ± 15.5	87.6 ± 19.1	87.1 ± 18.1	87.4 ± 19.9
T2 sound later	91.1 ± 13.7	84.1 ± 15.7	87.1 ± 18	89 ± 16.8
Consistent with T2	90.9 ± 13.9	86.6 ± 18.7	83.6 ± 18.2	85.8 ± 19.1

**Table 3 brainsci-13-00340-t003:** Descriptive statistics of Experiment 3.

	T1 (Mean ± SD) (%)	T2T1 (Mean ± SD) (%)		
		Lag1	Lag3	Lag8
T2 sound earlier	94.7 ± 6.6	9.35 ± 8.6	91.7 ± 8.4	88.4 ± 15.1
T2 sound later	94.6 ± 6.5	91.9 ± 7.2	91.8 ± 10.5	88.9 ± 15.8
Consistent with T2	95.3 ± 5.4	92.8 ± 7.9	91.3 ± 10.1	90.4 ± 13.8

## Data Availability

Data sharing not applicable.

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
