# Peer review of "Effect of Target Semantic Consistency in Different Sequence Positions and Processing Modes on T2 Recognition: Integration and Suppression Based on Cross-Modal Processing"

_brainsci, 2023, doi:10.3390/brainsci13020340_

Round 1
Reviewer 1 Report
The current manuscript reports some interesting results supporting the idea that cross-modal processing is influenced by top-down mechanisms. Authors performed three experiments to test whether attentional blink phenomenon in a cross-modal paradigm is modulated by top-down processing. Authors manipulated semantic coherence, the consistency between location of sound sequences and target 2, and information modalities. Although the content of the manuscript is generally well structured, it will benefit from language editing. Besides, there are some aspects that authors should address before the paper is ready for publication:
Regarding the hypotheses, a further specification of the expected results in relation to the theoretical explanation they preferentially support would be needed.
In the participants section authors should mention the recruitment procedure.
In relation to the description of experimental tasks procedure, I suggest authors to try to be more concise and clear. For example, in lines 134 - 135 it is not clear which letters were considered as confusing and which others were finally used as T1.
There is no mention to ethic aproval.
Concerning the results section, authors should provide descriptive statistics for all the measures at the different conditions, including kurtosis and skewness.
Moving to the discussion section, authors should re-think how to organize this section. Focussing discussion in answering the research questions instead of independently discuss results from the different experiments performed might be an alternative. Besides, authors should go deeper in discussing the limitations of this study and propose future lines of research.
Some minor issues (I would recommend authors to thoughtfully check throughout the hole manuscript, here some examples):
Line 122: I think authors did not mean that authors were paid according to Declaration of Helsinki, but that they followed the ethical principles of the Declaration of Helsinki in the design of their research. Authors should mention how much they paid to participants instead.
Line 147: the sentence seems incomplete.
Line 590: I'm not sure that sexual is the appropiate term in this sentence.
Figure 3 captions refer to experiment 4, but it seems that authors only report 3 experiments.
The p for p value should be in lowercase format following APA style.
Reviewer 2 Report
The manuscript reports results from 3 experiments that investigated semantic consistency of a target under different sequence position and presentation modalities (visual & auditory). The study found integration and suppression based on cross-modal processing of the second target.
I personally found very difficult following the results discussion in all 3 experiments and as a result did not see clear support for the conclusions. I believe a thorough revision of introduction, discussion, and results sections in all three experiments are needed before I can recommend that manuscript for publication. See my specific suggestions below:
Introduction:
P1, line 29 – please add a description of how attention blink would look like/be defined behaviorally.
The inhibition model (Raymond, 2000) – p.14, line468, interference model (Oberauer & Lin, 2017) - p.14, line475, and Gate model (Block & gruber, 2014) - p.16, line561 should be described in Introduction and for each of the experiments specific predictions of what pattern of result support which model should be added then the discussion of these models in the conclusion section could be shortened.
For all 3 experiments please remove detailed reporting of null results that is very confusing to the reader and takes attention away from the central message of the experiment, instead focus on the discussion of the significant findings and which of the claims in the conclusion these results prompted/supported.
Reviewer 3 Report
-
The writing does not meet the standards required by academic writing. Fragments could be found literally in every paragraph. The semantics of the writing could not be easily understood. These issues further lead to the low readability of the manuscript. In short, the writing/clarity of the manuscript should be largely improved before it could be further evaluated.
-
The authors must provide what lag1, lag3 and lag8 refer to.
-
The authors are suggested to explain why the length of fixation is not set consistently across different trials.
-
Are the assumptions for running ANOVAs made? If not, linear mixed models are generally used nowadays. Also, the authors run many statistical tests for one single experiment, how do the authors protect the increasing risk of Type 1 error?
-
Are all the responses included in the analysis? Data from participants whose accuracy rates are below a certain level should be eliminated. Please kindly refer to Liu and Chen’s (2017) and the references therein.
-
The authors need to clearly define what semantics mean in this study? Could the participants clearly hear the sounds and clearly understand the semantics of the sounds? Based on the results from Shen (1990) and Kuo et al. (2007), the length of a monosyllabic word in Mandarin Chinese is usually longer than 100 m.s.. Moreover, the length of a monosyllabic word is rarely shorter than 100 even when the participants articulate with a fast speech rate. (See also comment 9. I assume that the audio stimuli were in Chinese.)
-
In the second experiment, the authors do not clearly define what “earlier and later” refer to. The authors are suggested to define how early (in m.s.) or how late (in m.s.) those audio stimuli are presented.
-
The format of the section title “4.1.3” needs to be adjusted.
-
What is the language background of the participant? Are they monolingual? What is the native language? What is the language (the pronunciation) of the audio stimuli used in the experiments?
References:
- Kuo, Y.-C., Xu, Y. & Yip, M. (2007). The phonetics and phonology of apparent cases of iterative tonal change in Standard Chinese. In Gussenhoven, C., Riad, T. (Eds.), Tones and Tunes Vol 2: Experimental Studies in Word and Sentence Prosody (pp. 211-237). Berlin: Mouton de Gruyter.
- Liu, C.-T. J., & Chen, L.-M. (2017). Processing conjunctive entailment of disjunction. Language and Linguistics, 18(2), 269-295.
- Shen, X. S. (1990). On Mandarin tone 4. Australian Journal of Linguistics, 10(1), 41-59.
Round 2
Reviewer 1 Report
Authors have revised the manuscript according to reviewers comments. I thank authors for considering all the suggestions. However, there is some aspects that remain to be addressed. Authors should discuss about the implications of their results and future research.
Reviewer 2 Report
The revisions improved the manuscript.
Reviewer 3 Report
I thank the authors for sincerely considering the comments I provided in the previous review. After reading the manuscript and the authors’ responses to the review comments, I still hesitate to offer an unreserved recommendation to its publication.
1. First, the writing of the manuscript still requires much improvement. I understand that the authors have invited an agency to modify the language issues. However, the way the information is organized does not meet the high standard required by a prestigious journal such as Brain Sciences. I provide two examples below, although instances such as these occur multiple times in the manuscript. First, the authors mentioned “gray background” twice in the same section (i.e., section 2.1.2). This shows that the authors might not have thoroughly read through the manuscript and make sure that the information and the writings are concise and in a reader-friendly fashion. Second, in the responses, the authors mentioned that the participants passed College English Test 4 of China. However, in the manuscript, the authors claimed that all the participants had passed the Chinese English Level 4 test. What is the subject language of a Chinese English test? Is it about translation?
2. In the responses, the authors claimed that the assumptions for running ANOVAs were made and they stated that the relevant information was added in methods and results section. However, if the assumptions for running two-way repeated-measures ANOVAS are made, the authors must show that the distribution of the dependent variable pass the normality test. In the current manuscript, the authors simply claimed, without providing supportive statistics, that all data satisfied the condition.
3. The authors did not provide how they protect the increasing risk of Type 1 error in their responses and did not adopt any further measures to avoid the risk in the study.
4. The reason that studies from Shen (1990) and Kuo et al. (2007) were mentioned was to show that by playing a sound for just 75 m.s., the participants could not clearly hear or process the incoming information. In fast speech, a syllable that was shorter than 75 m.s. rarely happened. It seems that the authors did not take care of this issue. Simply by citing and including the references in the text would might not help.
5. What does it mean by excluding all data beyond three standard deviations? Does it mean that all the data from the particular participant were eliminated? How many participants were left?
6. Why did the authors use one-way ANOVA when the single group design was used?
7. The meaning of the sentence is not clear: In addition, the subject population was involved in Experiment 3 on separate days.
